# Multi-Modal Song Mood Detection with Deep Learning [note 1]

**DOI:** 10.3390/s22031065

**Published:** 2022-01-29

**Authors:** Konstantinos Pyrovolakis, Paraskevi Tzouveli, Giorgos Stamou

**Affiliations:** School of Electrical and Computer Engineering, National Technical University of Athens, 15780 Athens, Greece; tpar@image.ntua.gr (P.T.); gstam@cs.ntua.gr (G.S.)

**Keywords:** mood classification, convolutional neural networks, BERT, natural language processing, digital signal processing, deep learning, transfer learning

## Abstract

The production and consumption of music in the contemporary era results in big data generation and creates new needs for automated and more effective management of these data. Automated music mood detection constitutes an active task in the field of MIR (Music Information Retrieval). The first approach to correlating music and mood was made in 1990 by Gordon Burner who researched the way that musical emotion affects marketing. In 2016, Lidy and Schiner trained a CNN for the task of genre and mood classification based on audio. In 2018, Delbouys et al. developed a multi-modal Deep Learning system combining CNN and LSTM architectures and concluded that multi-modal approaches overcome single channel models. This work will examine and compare single channel and multi-modal approaches for the task of music mood detection applying Deep Learning architectures. Our first approach tries to utilize the audio signal and the lyrics of a musical track separately, while the second approach applies a uniform multi-modal analysis to classify the given data into mood classes. The available data we will use to train and evaluate our models comes from the MoodyLyrics dataset, which includes 2000 song titles with labels from four mood classes, {happy, angry, sad, relaxed}. The result of this work leads to a uniform prediction of the mood that represents a music track and has usage in many applications.

## 1. Introduction

The terms music and emotion are two concepts that are strongly connected, from the very first moment that man invented music. Special scientific interest presents the way that music causes emotional arousal to listener. In general, a track of music will cause the same emotions in a set of listeners, although there are a lot of factors responsible for the way a track will be perceived and will differ from listener to listener. Listeners’ backgrounds and tastes in music are two of the most important factors that will determine the emotions felt listening to a music track, but are not the only ones. According to Hunter [1], the emotional state and environment of the listener when the musical stimulus is received also play an important role.

Automated music mood detection constitutes an active task in the field of Music Information Retrieval (MIR) the past few years. Since music is produced and consumed more and more every day there is a growing need to effectively and optimally process these huge volumes of music. This need directly triggers the scientific field of MIR, which is the interdisciplinary science of retrieving information from music. MIR has many applications such as music classification [2], music recommender systems [3], music source separation [4], instrument recognition [5] and even music generation [6]. In this work, we will focus only on the task of music classification and more specifically on the task of music mood classification, which is categorizing music into predefined mood/emotion categories.

A first approach on the task of mood detection was made in 1990 where Gordon Bruner [7] who researched the way that musical emotion effects marketing. Later, Zaanen and Kanters [8] developed a Machine Learning model for music track classification based on emotion, utilizing lyrics and the TF-IDF embedding method. For emotional classification of tracks, the researchers Tzanetakis [9] and Peeters [10] train Support Vector Machines focusing on audio features like Mel Frequency Cepstral Coefficients (MFCC) and deal with MIREX (Music Information Retrieval Evaluation eXchange) classification tasks.

More recent works recruit the benefits of Machine Learning and Deep learning to address the problem of mood detection. In 2016, Lidy and Schiner [11] trained Convolutional Neural Networks using audio features, such as Mel-Spectrograms and MFCCs. An impressing progress on task of mood detection is made by Agrawal et al. [12] who suggested a lyric only transformer-based approach model training a XLNet network [13]. For the task of emotional classification, several multi-modal approaches were also made trying to utilize both the audio and lyrics of music tracks. A work worth mentioning is the work of Malheiro et al. [14] who constructed their own annotated dataset, extracted useful features and trained SVMs to predict the emotion evoked by music. On the other hand, Hu et al. [15] built a multi-modal mood classifier of a Hough forest using lyric features extracted from hierarchical DL models and saliency based audio features. Delbouys et al. [16] has also made important progress on the task, as a regression problem, predicting valence and arousal values by developing a multi-modal Deep Learning system as a combination of CNN, LSTM and Fully Connected layers. In our previous research paper [17] in agreement with all these works, we conclude to the same result, that multi-modal approaches overcome single channel models.

In this paper, inspired by the above, we further investigate how to detect the music mood applying a multi-modal analysis based on lyrics and audio signal. Firstly we analyze the lyric and then the audio applying deep learning techniques (CNN, LSTM, MLP, Tranformers). Then, we compare the results, applying a multi-modal analysis. The available dataset that we used for training and evaluation comes from the MoodyLyrics Dataset [18] which consists of a set of 2000 song titles alongside with their corresponding mood label, from a set of four basic moods—{happy, angry, sad, relaxed}. Subsequently, recommended architectures and data representations analyzed, and the experimental process for training and evaluation described in detail. The contribution of our research work in the field of music mood detection can be summarized in three key points: (a) multi-modal approaches are way more effective than uni-modal; (b) Transfer Learning and transformers can enhance the robustness of multi-modal systems; and (c) the correct extraction and combination of audio features can further improve the prediction goal.

## 2. From Audio and Lyrics to Mood

The ability to perceive emotion from music is said to develop from the first years of human life and evolves over time. Beyond age, a very important role in the perception of emotion from music have the cultural influences, as the study of Susino and Schubert suggests [19]. Therefore it is reasonable to say that the background and experiences of individuals determine their perception and interpretation of music, although in most cases evoked emotions from listening to music have a global interpretation [1]. In addition to listener features, music-related emotions also depend on other factors such as the influence of others or the environment music is listened to.

Natural Language Processing and Deep Learning will be applied to our task in terms of Emotion Recognition (Mood Classification). Emotion Recognition is the process that focuses on recognizing the emotion expressed by a human or identifying the emotion a medium (e.g., text, video, audio) can evoke to a human. However, we will mainly focus on the second clause of the definition. Since we have access in certain media, such as lyrics (text) and audio, we will try to recognize what emotions these media can cause in a listener. The general goal is to be able to develop a multi-modal emotion recognition system that will utilize text and audio concurrently and will be able to decide about the evoked emotion. The mood classification from audio and the corresponding lyrics are the base problems that we faced efficiently, utilizing text and audio features.

### 2.1. From Audio to Mood

Many studies research the nature of emotions caused by listening to a musical composition and which features of audio are responsible for emotion [20,21].

The structural features of music that contribute to the perception of musical expression are divided in two categories—segmental features and suprasegmental features. Segmental features are the acoustic structures that composite music, such as duration, tonal pitch and amplitude. Suprasegmental features are the individual features that determine a music track, such as tempo or melody. Some of the structural features of music are connected directly with the expression of certain emotions to listener [22]. Below is presented a table of correlation between structural features of music and emotions (Table 1).

In order to achieve an effective audio analysis, we have to adopt some methods [24,25,26,27,28] from the field of Digital Signal Processing, which make feasible the extraction of features with the desired information. To extract information related to emotion from the audio signal of a musical track, we used the following features:*Spectrogram* comes from the Short-Time Fourier Transform of a signal and expresses the sinusoidal frequency and phase content of signal’s sections (windows) as they vary over time. In practice, to calculate the STFT the signal is divided into segments of equal length to which the Fourier transform is applied, highlighting thus the Fourier spectrum. The process of displaying variable spectra as a function of time is known as a Spectrogram.In the case of continuous time, the function we want to transform, let x(t), is multiplied by a window function which is non-zero for a short time. Usually for the window function, w(t), the Hann or Gaussian type windows are used. The Fourier transform, for STFT, is calculated as the window slides up in the signal and mathematically calculated from Equation (Equation 1).
(1)$$STFT\left\{x\left(t\right)\right\}\equiv X\left(\tau ,w\right)=\int_{-\infty }^{+\infty }x\left(t\right)w\left(t-\tau \right)e^{-it}dt$$*Mel Spectrogram* is almost like *Spectrogram*, with the only difference that frequencies are transformed to the Mel scale. Mel scale is a scale of pitches, able to mimic the auditory system of a human.
(2)$$mel\left(f\right)=\frac{1000}{\log_{10}2}\log_{10}\left(1+\frac{f}{1000}\right).$$Mel filters mimic the response of the human auditory system with better performance than linear frequency bands. Essentially, the term Mel refers to a pitch scale created by experiments on listeners for the purpose of recognizing tonal changes perceived by the human ear. The rendering of the name is due to Stevens, Volkmann, Newman [24] and comes from the word melody (melody). *Log-Mel Spectrogram* is the *Mel Spectrogram* with a logarithmic transformation on the frequency axis.*Mel-Frequency Cepstral Coefficients (MFCCs)* comes from the *Log-Mel Spectrogram* with a linear cosine transformation.
(3)$$MFCC=\sqrt{\frac{2}{M}}\sum_{m=1}^{M}X_{m}\left(i\right)\cos\left(\frac{c\pi (m-\frac{1}{2})}{M_{m}}\right),$$
where $X_{m}$ is the logarithmic energy of *m*-th Log-Mel Spectrogram and *c* is the index of the cepstral coefficient.*Chroma* features, or pitch class profiles, are strongly correlated with music harmony and are widely used in music information retrieval tasks. Chroma features are not affected by changes in tonal quality (timbre) and are directly related to musical harmony. According to Müller [25], *Chroma* features are strong mid level features capable of retrieving important information from audio. If we assume the tonal scale used in western music (Equation 4), then it is easy to describe the matching between audio signal and chroma features. In practice, first we compute the *Spectrogram* of the signal and then for each window a vector is calculated $x=[x_{1},x_{2},...,x_{12}]$, where each element $x_{i}$ represents the corresponding element in scale (Equation 4).
(4)$$\{C,C^{\#},D,D^{\#},E,F,F^{\#},G,G^{\#},A,A^{\#},B\}.$$*Tonnetz* (or centroid tonal features) [29], is a pitch representation of signal. Vector $t_{n}$ (see Equation (Equation 5)) is the result of multiplication *chroma* vector (see Equation (Equation 4)) and a transformation matrix *T*. Thereafter, vector $t_{n}$ is divided with the $L_{1}$-norm of vector $c_{n}$.
(5)$$t_{n}\left(d\right)=\frac{1}{∥c_{n}{∥}_{1}}\sum_{i=1}^{11}T\left(d,l\right)c_{n}\left(l\right),$$
where d∈[0,5] refers to the index of the element calculated and l∈[0,11] is referred to the index in *chroma* vector.*Spectral Contrast* features represent the intensity and contrast of spectral peaks and valleys. To compute these features, first, we compute the *Spectrogram* of the audio signal and the result is fed to an octavian scale filter. Then, a Karhunen–Loeve transformation is applied to map the features to a rectangular space and eliminate information on random directions. The result of this process is a powerful representation of the peaks and valleys of the spectrum, and the contrasts between them can be easily identified with these features.

The extraction of these features is performed so we can later experiment with their different combinations and decide which features contain information applicable to the task. Figure 1 (mel spectrogram, log-mel spectrogram, MFCC) and Figure 2 (chroma, tonal centroids, spectral contrast) represent these extracted features of a sampled track (see Figure 3—audio signal, Figure 4—db spectrogram) from the available dataset. The song is *Sabbath Bloody Sabbath* by *Black Sabbath*, a classic heavy metal song and its corresponding label is ‘angry’. The sample song has both instrumental and vocal parts and it is a mixed production, compressed in mp3 file format.

### 2.2. From Lyrics to Mood

Natural language processing (NLP) employs computational and linguistics techniques to understand human languages in the form of text and speech/voice. Prominent contributions to the field of NLP under active research include sentiment analysis with mood detection such as happy, sad, angry, and so forth. Mood classification has been studied from two points of view. The first one is dealing with emotions as discrete units, while the second point of view is classifying emotions in groups, in a dimensional space. Dimensional models try to interpret human emotions with vectors from spaces of two or three dimensions.

A powerful dimensional model, which we used in our work, is the model Circumplex developed in 1980 by James Russel [30]. According to Circumplex, all human emotions are distributed in a circular two-dimensional space with axes of valence and arousal (see Figure 5). Russel supports that any human emotion, unprecedented or not, can be represented in this two-dimensional space by a pair of values.

This section deals with the first part of a multi-modal emotion recognition system, that is the NLP subsystem, which will use text information as input data. The text information available comes in the form of lyrics. Lyrics are words that make up a song and contain strong and meaningful information about the emotional state a song can evoke to a listener. Trying to mimic the effect lyrics have to a listener we will build a NLP/Deep Learning system that will predict the emotion, the textual part (lyrics) of a song, can cause in a listener.

#### 2.2.1. Word Embeddings

However, in NLP models, words that constitute a corpus, in our case lyrics, do not show any direct numeric information. In order lyrics to have meaning, for the models we developed, we had to represent each word with a vector, these vectors are known as Word Embeddings. To compute these vectors we followed many known methodologies, such as Bag of Words (BoW) [31], TF-IDF [32], Word2Vec [33], GloVe [34], BERT embeddings [35] which produced vectors of different representations and dimensions. The model we will mainly use for lyric analysis, BERT, takes as input sequences of text and transforms them internally to tensors of size (3, 128). BERT’s tokenizer is responsible for this process and it is described in more detail subsequently.


*Bag of Words*


The Bag of Words (BoW) method [31] is the simplest text to vector technique, describing the occurrence of words within a document. Each text is represented from the set (bag) of words. Specifically, given a vocabulary $V=[v_{1},v_{2},...,v_{v}]$, where |V|=v is the number of words of the text *C*, where |C|=c≤v. The BoW algorithm constructs a vector $w=[w_{1},w_{2},...,w_{v}]$ with *v* length; element $w_{i}$ of the vector expresses how many times the word $v_{i}$ is in the text *C*. Having a vocabulary, for example, V=[today,yesterday,my,dog,cat,a] and a text document, for example, *C = ‘Yesterday my dog met a dog’*, then the generated vector will be w=[0,1,1,1,2,0,1]. Each BoW involves a vocabulary of known words and a score of the presence of known words. The model is only concerned with whether known words occur in the document.

In this approach, we look at the histogram of the words within the lyrics document, i.e., considering each word count as a feature. The intuition is that lyrics documents are similar if they have similar content. Further, that from the content alone, we can learn something about the meaning of the document. The complexity of this algorithm comes both from deciding how to design the vocabulary of known words (or tokens) and how to score the presence of known words.

For many years, the Bag of Words algorithm was used with great success in many problems in the field of natural language processing. This algorithm with small variations is still used today. On the other hand, the BoW method has several disadvantages as the only information that it retains is the number of occurrences of words. The most important disadvantage is that all information about the syntactic structure of the text is lost. An example of this problem is the sentences ‘Make love not war’ and ‘Make war not love"—the algorithm will give exactly the same vector for their representation, although the their meaning is completely different


*TF-IDF*


The TF-IDF method [32] is a statistical method that calculates the relevance of a word in a text from a set of texts. This value depends on two quantities: the frequency of the term Term Frequency (TF) and the Inverse Document Frequency (IDF) of the documents. The TF value is calculated in several ways, with the simplest one being the frequency with which a term, *t*, appears in a document, *d*. The IDF expresses the frequency of the word in all the documents and is calculated as the logarithm of the ratio of the total number of documents to the number of documents in which the word appears.

The value TF-IDF for a word *w* in a text *d* from a set of texts *D* is calculated by the Formula (Equation 8), where values close to 0 mean the significance of the word in the text is small, while values close to 1 indicate great importance.
(6)TF(w,d)=freq(w,d)
(7)$$IDF\left(w,d,D\right)=\log\left(\frac{N}{count(d\in D:w\in d)}\right),$$
(8)TF−IDF(w,d,D)=TF(w,d)·IDF(w,d,D).

The examination of the significance of words over a large set of documents is conceded as the main advantage of the TD-IDF method. For example, words that often appear in a text, such as ${}^{‘}{\mathrm{is}}^{’}{,}^{‘}{\mathrm{what}}^{’}{,}^{‘}{\mathrm{the}}^{’}$, are not important to the semantics of a document. The algorithm manages to recognize these words and give them small importance. Instead, words like ${}^{‘}{\mathrm{bug}}^{’}$, for example, could appear frequently in a document about insects or debugging code. In these cases, the algorithm can detect these words, giving significant meaning. The drawback of this method is the ignorance of the text’s syntactic structure, (like the BoW method does). In addition, the effectiveness of this method is dependent on the large number of documents that the set of texts (corpus) contains.


*Word2Vec*


The term Word2Vec [33,36] refers to a set of models, used for the vector representation of words that a text contains. A Word2Vec model receives as input a large volume of documents (corpus) and produces a vector space $R^{N}$, typically N>100. Each unique word belonging to the corpus is represented in the vector space by a vector, in such a way that words with similar contexts are close together. In this way, the correlation of the words that have similar conceptual meaning is achieved.

In the Word2Vec models, the word vectors are presented using the one-hot encoding, which means that each vector has a length of *V*, and the vocabulary size consists of zeros in all its elements except from the item that represents that word in vocabulary, which takes the value 1. These words vectors are inputs in the neural network, where in its hidden layers the input products are summed with the parameter tables and at the output layer the softmax function is applied predicting the correct positions of ones and, consequently, the correct words in the one-hot vectors of the output.

For the vector representation of the words, these models choose between two architectural modelings, the *CBoW* and the *Skip-gram*. The *CBoW-Continuous Bag of Words* model takes as input the neighboring words of a target word and tries to predict the target word in the output. In contrast, the *Skip-gram* model accepts a word as input and predicts its neighboring words as output. The Skip-gram model works best for small volumes of data and manages to represent rare words better. On the other hand, the CBoW model better represents more common words and it is faster.


*GloVe*


The GloVe (Global Vectors) method [34,37], similarly to Word2Vec, uses vector word representation. The advantage of this method is that it not only uses local attributes, such as contextual words, but also incorporates some global attributes. The model is capable of recognizing words that, although appearing in the context of another word, do not convey any conceptual content. For example, in the sentence *‘The dog chased the cat’* the word *‘the’* precedes the words *‘dog’* and *‘cat’* but does not give them any information which must also be recognized by an NLP system.

Firstly, the GloVe model creates a vector space, utilizing a set of documents. It achieves this by calculating and utilizing the co-occurrence of words in a volume of documents (corpus). The key component of the method is the *C* co-occurence table *V* × *V* for a *V* size vocabulary, where each element $C_{ij}$ of the table expresses the probabilities that the word $w_{j}$ occurs in the context of the word $w_{i}$.

Then, the GloVe method utilizes the co-occurrence table and calculates ratios from the co-occurrence values of two words to discover their conceptual relationship. Suppose that P(k|w) expresses the co-occurrence, that is, the probability that the word *k* belongs to the context of the word *w*. For example, if we have two words $k_{1}=ice$ and $k_{2}=steam$, then the ratio P(solid|ice)/P(solid|steam) would be large because the term P(solid|ice) would have a price close to 1 while the term P(solid|steam) price close to 0. Respectively, the ratio P(gas|ice)/P(gas|steam) would have a very low value, while the ratio P(water|ice)/P(water|steam) would have a value close to 1.

The procedure that the GloVe method follows to predict contextual words is to use the probabilistic co-occurrence ratios by performing regression. In conclusion, GloVe generates a *f* function that takes a word as an argument and generates a vector of that word. The characteristic of these vectors is that adding and subtracting them corresponds not only to the vector space but also to the conceptual space. Such an example being the proposition: $f{(}^{\prime }queen^{\prime })=f{(}^{\prime }king^{\prime })-f{(}^{\prime }man^{\prime })+f{(}^{\prime }woman^{\prime })$.


*Bert embeddings*


The model we will mainly use for lyric analysis, BERT, takes as input sequences of text and transforms them internally to tensors of size (3, 128). BERT’s tokenizer is responsible for this process and it is described in more detail subsequently.

The BERT model requires a specific representation of input data in order to work correctly, called BERT Embeddings [35]. This representation requires the breaking of the input sequences in tokens, BERT provides a custom Tokenizer for this process. Specifically, BERT’s Tokenizer creates a sequence of word-tokens, matching each input word to BERT’s dictionary. When tokenization is complete, Tokenizer puts the token [CLS] at the beginning and the token [SEP] at the end of each sentence. BERT expects three parallel vectors for each input, these vectors have fixed length of 128 and have the names *input_ids*, *input_masks and segment_ids*. Vector *input_ids* is constructed from identification IDs of each token in input sequence, these IDs are provided in model’s dictionary. Vector *input_masks* is constructed with ones in the first *n* items and zeros in the last 128−n items, where n is the length of input sequence. Last, vector *segment_ids* helps to separate sentences that construct an input sequence, for example, the items that match the first sentence will be zeros, the items that match the second sentence will be ones; the items that match the third sentence will be twos and so forth. Input sequences longer than 128 will be truncated and input sequences shorter than 128 will be filled with empty tokens.

## 3. Dataset Preparation

Mood detection, in the context of this paper, has been approached from two different directions. The first one is associated with lyric analysis while the second concerns the analysis of the audio signal of a music track. The creation of the dataset is based on a common corpus (MoodyLyrics) which we used for lyric and audio signal analysis. This dataset provides limited information such as the song titles, corresponding artists’ names and mood labels, and it does not contain the desired data (audio files, lyrics) or any information about the genre. The reason we choose to work and experiment with this dataset was the fact that, to our knowledge, it was the biggest data source that could provide mood class labels for music songs, which was essential to our target goal, the classification task.

### 3.1. Lyrics Dataset Corpus, Features and Preprocessing

The available dataset we used comes from the MoodyLyrics Dataset [18] which consists of a set of 2000 song titles alongside with their corresponding mood label, from a set of four moods {happy, angry, sad, relaxed} (see Figure 5). The process of song classification is based on Russel’s emotional model.

According to [18], the first process for creating this dataset is related to lyrics collection. Later, each word in lyrics is attributed to a pair with values of valence and arousal. The set of values for each word is computed with the help of three dictionaries which contain emotional information, ANEW (Affective Norms of English Words) [38], WordNet [39] and WordNet-Affect [40]. Subsequently, for each song a general pair of values of valence and arousal is computed by the individual values of the words that make up the lyrics of the song. Finally, each song is addressed to one mood label by discriminating each song to one of four mood classes, according to Table 2.

Alongside the mood label for each song, MoodyLyrics Dataset contains information about the title of the song, the artist and an identification string (ID). These information were used to download required data from the web.

To achieve better performance of our machine learning systems and augmentation of the dataset, each lyric text was divided to subtexts of four lines. After this process we ended up with 18,115 elements in our dataset instead of 2000.

#### Proposed Words Embeddings

For our corpus (songs’ lyrics), we developed models for represent each word with a vector (Words Embedding). To compute these vectors we followed the methodologies that are described in (Section 2.2.1). The proposed word embedding for each of these models will be presented in the following.


*Bag of Words*
The corpus preprocessing included the replacement of all the uppercase characters of the English alphabet with the corresponding lowercase characters, the removal of all punctuation and English stopwords (e.g., “the”,“is”,“at”,“which” …). Then, the preprocessing continues with the stemming procedure to reduce the derived words to their base stems, the tokenization of the input text in tokens and the rendering of IDs in the terms. After embedding the input data with the BoW method, our vocabulary size is |V|=21,266. From the vocabulary, the 10,000 most common words are chosen to be used for the representation of the texts. Thus, each quatrain is represented by a vector of 10,000 in length, where each element corresponds with the number of occurrences of the corresponding vocabulary word in the quatrain. At the embedding layer layer of model $T_{1}^{\prime }$ the input vectors are compressed into dense vectors of length EMBEDDINGSIZE=128.
*TF-IDF*
The preprocessing in the TF-IDF method is the same as in the BoW method. The vocabulary size is the same and the 10,000 words with the highest frequency terms are retained. This time, however, the words are represented by the resulting vectors according to the relation (Equation 8). Again, at the embedding layer, the input vectors are compressed into dense vectors of length EMBEDDINGSIZE= 128.
*Word2Vec*
Before the Word2Vec method, a preprocessing takes place, converting all characters to lowercase and removing punctuation, numeric characters and stopwords. The processed input sequences are used to train the model. The model produces vectors of length |s|=408, representing the sentences of each sequence. The value |s| is the maximum number of words in the input sequences, each $s_{i}$ element represents one word in the input sequence, and the remaining elements are replaced by zeros.At the embedding layer its dimension is set to EMBEDDINGSIZE=128 while this time the embedding initializer is also defined as the table, dimension |V|xEMBED−DINGSIZE, which consists of the integrations of all of vocabulary words and converts each input word to the corresponding vector (embedding vector).
*GloVe*
The GloVe method uses the same procedure as Word2Vec to preprocess the input sequences. For word embedding, GloVe provides default pretrained representations. It is essentially a 400,000-word vocabulary with pretrained representation vectors of length 100. From these vectors, the 400,000 × 100 co-occurrence matrix is constructed, which will then replace its untrained weights in the level of integration. Because the GloVe model uses finite vocabulary, it ignores words outside the vocabulary and it replaces their vectors with zeros.
*BERT Embeddings*
BERT’s tokenizer on the other side does not require any further preprocessing from our side. The mechanism (tokenizer) BERT uses is responsible for all preprocessing taking place in the input sentences, before training the model.Specifically, the tokenizer accepts at its input the texts for tokenization and creates a sequence of terms-words matching each input word in the corresponding term provided by its vocabulary, while maintaining occurrence order of words. For input words that are not recognized in the vocabulary, the tokenizer will try to break them down in vocabulary tokens to the maximum number of characters, in the worst case it will split a word into the characters that make it up. In the case of splitting a word the first token will appear in the sequence as it is, while the other tokens will appear augmented by the double symbol # at the beginning, so the model can identify which tokens are the results of splitting. For example, the word “playing” will be split into tokens “*play*” and “*##ing*”.

### 3.2. Audio Dataset Corpus, Features and Preprocessing

In order to apply dataset augmentation to the audio signal data, we divided audio files in 10 s clips and the result was 37,989 audio clips with 10 s duration. Alongside the fragmentation of audio files, downsampling of audio signals were applied, the sampling rate of audio decreased to 22,050 Hz from 44,100 Hz. We observed that a high sampling rate does not play an important role in mood detection, but its decrease helped significantly with the dimensional reduction of the input data. The bitrate of the available songs varies from 96 kbps to 256 kbps, but that deviation does not have any important impact to our work since bitrate does not have an important effect on the evoked emotion. To create the audio dataset, we use string pairs (title, artist) to collect audio files from the web, in mp3 format and mono channel mode.

The audio files, however, are not in a format that can be used immediately for the training of the systems developed. To extract features from audio files we used the popular, for MIR tasks, python library librosa https://librosa.github.io (accessed on 15 December 2021). From librosa library, we specifically used the functions melspectrogram(), mfcc(), chroma_stft(), tonnetz(), spectral_contrast() to extract these features. The above process has, as a result, the construction of input tensors of size (60, 431, 6), for each 10 second audio clip, which will be later fed to the CNN and fusion models.

## 4. Methods

System development, in the context of our work, will be accomplished separately for text and signal. The combination of these two systems into one uniform system that utilizes both text and signal features, presents a better performance. It is scientifically known that emotional arousal caused by a piece of music is due to both the lyric synthesis and the music composition. So our goal from the beginning of this paper was the utilization of both features in the development of a multi-modal deep learning system [41,42].

### 4.1. Lyric Analysis Subsystem

For lyric analysis we use three different architecture approaches, one Fully-Connected Neural Network ($T_{1}^{\prime }$), one Recurrent Neural Network ($T_{1}$) with LSTM cells [43] and one pretrained Network ($T_{2}$) that utilizes Transformers, also known as BERT (Bidirectional Encoder Representations from Transformers) [44]. For each architecture we use different word embedding methods: for the Fully-Connected Network we use Bag of Words and TF-IDF embeddings, for the LSTM Network, which utilizes the words’ order of appearance, we use Word2Vec and GloVe embeddings and for BERT we use BERT embeddings.

#### 4.1.1. LSTM-MLP Models

Due to their strong characteristics, LSTM networks ($T_{1}$) were selected in the framework of this work, as a first attempt to build a neural network for analyzing emotion from text. In particular, the network consists of one layer of integration of representation vectors, one layer of LSTM cells and two layers of fully connected neurons. However, the vectors produced by Bag of Words and TF-IDF text embedding methodologies do not have a sequential information, therefore for training with these two embedding methodologies we will subtract the layer of LSTM neurons, this modified architecture will be referred to as $T_{1}^{\prime }$ (Figure 6). From an architectural point of view, the network’s hyperparameters were selected after application of grid search, a technique used for optimizing the parameters of the network. Specifically, the dimension of the first layer of integration were of variable size and were selected according to the embedding technique applied in each case.

For the construction of the second layer, 64 LSTM neurons were selected with tanh as the activation function, sigmoid as the recurrent activation function, dropout rate equal to 0.2 and recurrent dropout rate equal to 0.2. Then the second layer, of fully connected neurons, constructed from 128 neurons, with zero dropout rate and the activation function the ReLU function. The third and final layer consists of four fully connected neurons with the softmax function as the activation function. Essentially, the last two layers of fully connected neurons compose a classifier, responsible for classifying the samples into the four mood classes.

#### 4.1.2. BERT Model

The continuous progress that transformers make in NLP tasks in recent years made BERT the architecture of choice for the task of emotional analysis of lyrics. The power of transformers is due to their bidirectional training technique combined with Transfer Learning’s ability to reuse the knowledge acquired from a task to another. BERT’s ability to reuse the knowledge exported from a large corpus of English texts make it a top choice for the development of a NLP system like ours. In our system, we chose to use the pretrained model BERT-base uncased, which consists of 12 layers with a hidden layer size equal to 768 and 110 million total parameters.

Each of the 12 layers of BERT is actually an encoder, each encoder consists of three different processing layers of embedding vectors. The first layer implements an attention mechanism called multihead-attention with 12 heads of attention. The second layer consists of a normalization layer and a feedforward network and the last layer of each encoder is a position encoding mechanism that inserts position information in embedding vectors.

On top of BERT, we placed a classifier of two fully-connected layers of neurons. The first layer consist of 256 neurons with activation function the sigmoid function and dropout rate equal to 0.1. The second layer consists of four neurons with activation function the softmax function.

### 4.2. Audio Analysis Subsystem

For audio signal analysis we developed a Convolutional Neural Network ($A_{1}$). The ability of CNNs to perceive and utilize time and space dependencies of 2D and 3D vectors and the parallel processing they offer, made them the architecture we chose to develop the audio signal model $A_{1}$.

To organize the architecture of the CNN we developed, we constructed three pairs of convolutional layers (Conv2D) and pooling layers followed by a classifier of two fully-connected layers. Network hyperparameters were adjusted with the help of grid search method, a technique that highlights optimal hyperparameters from a set of values, testing all possible combinations and selecting the combination that achieves optimal system performance.

Specifically, for the first convolutional layer 128 different filters were chosen, with kernel size of 6 × 6 and the function ReLU as its activation function. The first pooling layer, which follows, implemented the method max pooling with window dimensions of 2 × 2.

The second convolutional layer uses 256 filters has a 5 × 5 kernel and activation function the ReLU function. In this layer we used a padding technique so the dimension of the layer’s output would be the same as the input’s. The second pooling layer is identical to the first. The third convolutional layer uses 512 filters, has a 4 × 4 kernel, an activation function as the ReLU function and implements padding on its input. The third pooling layer is again the same as the previous two.

Before the classifier, we intervene a flattened layer, which transforms three dimensional tensors to one dimensional tensors. The first layer of the classifier consist of 16 neurons with activation function the ReLU function and dropout rate equal to 0.6. The second layer consists of four neurons with an activation function as the softmax function, in order to calculate the classification probabilities of inputs into the four mood classes of the output.

In order to avoid over-fitting in this network, normalization and regularization were applied. The regularization techniques are functions that apply penalties to the parameters of a layer during training. Essentially the normalization penalties apply to a cost function trying to optimize the network’s parameters. In the case of network $A_{1}$ the function L2 will be applied as a normalization function. The L2 regularization function in practice adds the square of the parameter’s value as a penalty term in the cost function.

For example, if the cost function is the sum of the squares (Equation (Equation 9)), then normalization L2 will add to the cost function the term shown in Equation (Equation 10). When λ is zero then we return to Equation (Equation 9), otherwise when λ is too large it will add a large penalty and may lead to under-fitting the data. For this reason the choice of λ must be very carefully examined—in our network we conclude to the optimal values after grid-search. L2 regularization is applied to the the convolutional layers of $A_{1}$ network with λ=0.00001.
(9)$$L\left(x,y\right)=\sum_{i=1}^{n}{\left(y_{i}-\sum_{j=1}^{p}x_{ij}\beta_{j}\right)}^{2}$$
(10)$$L\left(x,y\right)=\sum_{i=1}^{n}{\left(y_{i}-\sum_{j=1}^{p}x_{ij}\beta_{j}\right)}^{2}+\lambda \sum_{j=1}^{p}\beta_{j}^{2}.$$

A data normalization process will also be applied to the first fully connected layer. Batch normalization [45] is a technique for improving the speed, performance and stability of Artificial Neural Networks. To achieve the above, in practice this technique normalizes the output of an activation layer by subtracting the average value of the batch and dividing by the batch’s standard deviation. In this way the average value of the activation is maintained close to 0 and the standard deviation of the activation close to 1. In this way extreme values that would be cut off from training process are now normalized to more normal values and utilized in the training. Let *B* denote a batch of data and $\mu_{B}$, $\sigma_{B}$ are the corresponding mean and variance of B, of size *m*. Batch normalization will transform an input vector $x=(x^{\left(1\right)},...,x^{\left(d\right)})$ to its normalized version $\hat{x}$, as Equation (Equation 11) suggests.
(11)$${\hat{x}}_{i}^{\left(k\right)}=\frac{x_{i}^{\left(k\right)}-\mu_{B}^{\left(k\right)}}{\sqrt{\sigma_{B}^{{\left(k\right)}^{2}}+\epsilon }}$$
where k∈[1,d],i∈[1,m] and ϵ is a small constant that adds stability.

To feed the CNN with data, first we follow the process described in Section 2 for feature extraction. Feature extraction method produces six feature matrices of size 60 × 431 for each audio clip. These matrices will be stacked together to feed the network; later we will study the best possible combination and stacking of these matrices. For the STFT, in feature extraction, we used a Hanning window, FFT window size equal to 1024 and step length equal to 512.

### 4.3. Fuse Analysis System

Our final approach on the task of mood detection of music tracks concerned the development of a uniform system that would utilize both text (lyric) features and audio features. As evidenced empirically, the emotional excitement evoked in the listener of a piece of music, is due both to the composition of the lyrics as well as the composition of the music. Thus, the goal of this work was from the beginning the exploitation and analysis of these characteristics for the development of a Machine Learning system. For the development of this uniform system, the previously described systems that achieve the best performance in both text and audio signal analysis will be used.

The development of this system ($M_{1}$) Figure 7 will be implemented with the use of architectures BERT ($T_{2}$) and CNN ($A_{1}$), as described in this section. In practice we will merge the already trained systems $T_{2}$ and $A_{1}$ to a more complex and robust system. To merge these two systems together we will combine their outputs with a technique called late fusion. In practice, to combine the outputs we will feed them in one common classifier which will make one general prediction for each song, utilizing the knowledge of both systems. To construct the common classifier we used two fully-connected layers of neurons; the first layer is a combination of 16 neurons with the ReLU function as the activation function, while the second layer is a combination of four neurons with the softmax function as the activation function.

The problem that the multi-modal system raises is how we would combine the input data to make a general prediction. As we have already said to augment our dataset lyrics where split in sub-lyrics of four lines and audio were split into 10 s audio clips; the problem with these segments of data is that they are not aligned. To overcome this issue we decided to merge the segments that make up a song and make two general predictions for each song, one from text and one from audio, and then feed these predictions as inputs to the classifier. These general predictions for each song were computed by the arithmetic mean of predictions that models output for each segment of a song. Finally, two general predictions for each song are computed and fed to the classifier.

## 5. Experimental Results

This section will analyze the process of training and verification of the systems described in Section 4. Once the architectures and the way that data will be fed to the models are complete, the next step is to start the experimental and iterative process of monitoring their training and evaluation. In the experimental process 80% of the available data have be used as training data while the rest 20% as evaluation data. From training data 10% have be used as validation data. Additionally, to train our models we used Backpropagation as the training algorithm and Categorical Crossentropy as the cost function.

Figure 8 and Figure 9 display the distribution of available data in the four mood classes.

### 5.1. Lyric Analysis Subsystem

The experimental process for lyric analysis started with the simple models $T_{1}$ (LSTM) and $T_{1}^{\prime }$ (MLP), combined with several text embedding techniques (BoW, TF-IDF, Word2Vec, GloVE), and ends with the complex transfer learning model $T_{2}$. We will not get in much details about the models $T_{1}$ and $T_{1}^{\prime }$ since they were not the main models we experimented with. The available data for this task consist of 18.115 pairs of four-line lyrics and their mood labels. The performance of the training and evaluation of the model $T_{2}$, as well as the comparison of its performance and the other text models will be presented in this section.

To train model $T_{2}$ text was transformed in representations BERT expects, as described in Section 4.1. For the training process, we selected *Adam* as optimizer, with *learning rate* equal to 0.0005 and *beta${}_{1}$* and *beta${}_{2}$* parameters set to 0.9 and 0.999 respectively. *Number of epochs* for this process were chosen to be equal to 10 and *batch size* equal to 128.

BERT model comes already pretrained in the form, provided by Google. This model’s training method is called *fine-tuning* and focuses on the last layers and only adjusts their parameters, the first layers are left untouched. In our model fine-tuning affects only the two fully-connected layers of the classifier and only the last layer of BERT.

In Figure 10 and Figure 11, we represent *Accuracy* and *Loss* values for training and validation data. From these figures we can conclude, noticing test curve, that model $T_{2}$ performs badly on validation data. This performance problem is connected with the volume of the available data, BERT needs big volumes of data in order to perform to the maximum, while our dataset is quite small. In Table 3, we display evaluation metric values, on evaluation data, which model $T_{2}$ achieves compared with model $T_{1}$ and $T_{1}^{\prime }$.

Two basic reasons that lead to high levels of performance and finally in the choice of model $T_{2}$, are the complex attention mechanisms and the big volume of data that BERT utilizes for its pretraining. One more outcome of the experimental process is the improvement in performance and training time that arises after fragmentation of lyrics in four-line lyrics.

### 5.2. Audio Analysis Subsystem

The experimental process for audio analysis concerns the training and evaluation of the CNN model $A_{1}$. The available data for this task consist of 37,989 pairs of 10 s audio clips end their mood labels. In this section we will focus on the evaluation and training of the model $A_{1}$, we will compare its performance with the rest models and we will search for the best available feature combination.

The first task of the experimental process was to find the best possible combination of feature matrices, that result from the Digital Signal Processing methods we describe in Section 2. These matrices with size 60 × 431, are stacked together in different possible combinations and then are fed as inputs in the CNN. In Table 4, we display the results these combinations achieved. As we can see in the last row the combination of all six feature matrices is the one that achieves the best Accuracy percentage and consequently the combination of features we chose to train and evaluate the model $A_{1}$.

In the training process of model $A_{1}$, we chose *number of epochs* equal to 10 and *batch size* equal to 16. For this process, we selected *Adam* as an optimizer with *learning rate*, *beta${}_{1}$* and *beta${}_{2}$* parameters set to 0.001, 0.9 and 0.999, respectively. In Figure 12 and Figure 13, we represent *Accuracy* and *Loss* values for training and validation data. Finally, in Table 5, we display the evaluation metric values that model $A_{1}$ achieves compared with models $T_{1}$ and $T_{2}$. For $T_{1}$, we chose the embedding method TF-IDF because it is the one with the best results.

CNN’s nature allows the detection of areas in feature matrices that are responsible for valence and arousal. In Table 5 the audio model $A_{1}$ outperforms the text model $T_{2}$ in the context of music mood detection. Again, data fragmentation into 10 second audio clips leads to improvement in performance and training time.

### 5.3. Fuse Analysis System

To construct the fuse model $M_{1}$, we utilized the models $T_{2}$ and $A_{1}$, adding a classifier at the top. However, model’s $M_{1}$ training is limited in the adjustment of the synaptic weights and biases of the last layers that synthesize the classifier. Models $T_{2}$ and $A_{1}$ are used with fixed parameters and they do not require training.

Concerning the training process of the model $M_{1}$, we chose *number of epochs*, *batch size*, *learning rate*, *beta${}_{1}$* and *beta${}_{2}$* parameters set to 10, 16, 0.001, 0.9 and 0.999 respectively. In Figure 14 and Figure 15, we represent *Accuracy* and *Loss* values for training and validation data. Finally, in Table 6, we display the evaluation metric values that model $M_{1}$ achieves, compared with the previous models.

Fusing the two subsystems into one complex system achieves a huge improvement in performance and outperforms single channel systems, as is depicted in Table 6. This outcome is a result of the exploitation of both lyric and audio data and making one concentrated prediction for each song by its segments. Additionally, in Table 7 we compare our model (BERT + CNN) with other models that utilize the same dataset (MoodyLyrics) as us [12,46,47,48,49]. We should also mention that our final model outperforms previous approaches that use different sources of data, specifically our model achieves 94.32% *F*1 *Score* while Malheiro’s model [14] achieves 88.4% *F*1 *Score*.

All experiments were performed on a Linux server (5.4.0-80-generic Ubuntu) equipped with 64GB DDR4 memory, one 16-core Intel Xeon CPU (2.10GHz, 32 threads) and two NVIDIA GeForce GTX 1080 GPUs (8GB of DDR5 RAM). The programming language of our choice for experimentation was Python https://www.python.org (accessed on 15 December 2021) and our models were developed based on the deep learning libraries TensorFlow https://www.tensorflow.org (accessed on 15 December 2021) and Keras https://keras.io (accessed on 15 December 2021). The source code for the experimental process is available at [50].

## 6. Conclusions and Future Work

In this work, we approached the task of detecting musical mood by analyzing lyrics and acoustic signals, applying the principles and logic of supervised learning. Implementations of deep learning systems have been introduced in conjunction with some data representation methods based on natural language processing and digital signal processing techniques. The process was completed with the training and evaluation of the three proposed systems—lyrics only, audio only and multi-modal. The experimental process verifies the initial hypothesis; multi-modal systems are superior to uni-modal systems. In the context of detecting the mood evoked by music, both the lyrics and the audio contain useful information for training deep learning models. If the question to be answered was *which part of music is correlated the most with the emotion felt?*, according to our findings the audio part of music has more influence in terms of emotion. However, when both the lyrical and the audio part of a song is combined the results are astonishing—multi-modal systems overcome by far uni-modal systems with almost 24% improvement in accuracy.

One more important finding of our work is the importance of feature extraction from the audio signal files. By training and evaluating the developed CNN with multiple combinations of the extracted features, we can clearly assume the influence each feature has and which features are valuable for the task. On the other hand, training and evaluating a system that utilizes text (lyrics in our case) required experimentation with different ML/DL architectures. In this direction the trend of NLP could not be overlooked; transformers (BERT) live up to their reputation as being the architectures that produce the best results. Finally, one more important conclusion of our work is the effectiveness of Transfer Learning with the task. Both the NLP model and the final fusion model incorporate the concepts of TL in order to produce optimal and robust results. We hope our work will inspire the potential reader and will be a trigger for engaging in the field.

Future work can be accomplished in many and different directions. The first and probably the most important direction concerns the size of the dataset to be used for the effective development of a robust deep learning system. Because the data we used in this work was limited, future work would include developing a system that implements unsupervised learning from unlabeled data, as it is available in huge volumes. An even better solution for improvement would be the combination of large amounts of unlabeled and small amounts of labeled data to develop systems that would integrate the methodologies of semi-supervised and/or self-supervised learning. In terms of data, the use of data that has lyrics and audio aligned will also add great value and robustness to a multi-modal system like ours. Another direction for the future development of this work could be to better manage the audio signal and to identify an optimal combination of new exported audio and text features. For example, audio features that would represent the tempo and beat of a song could add great value to the task of mood classification. Furthermore, great added value will be a study of how transformers, combined with CNNs, could be used for the better utilization of audio-signals. Finally, we would love to, or inspire others to, apply the models and knowledge acquired from this work to more MIR classification tasks (genre classification, music tagging, artist classification) or in other multi-modal problems such as the task of emotion detection from video.

## Figures and Tables

**Figure 1 sensors-22-01065-f001:**
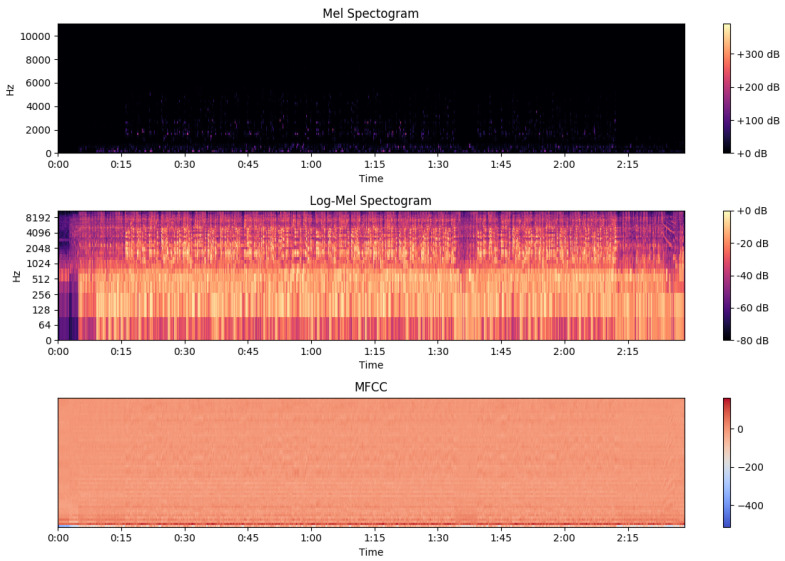
Mel Spectrogram, Log-Mel Spectrogram, MFCC.

**Figure 2 sensors-22-01065-f002:**
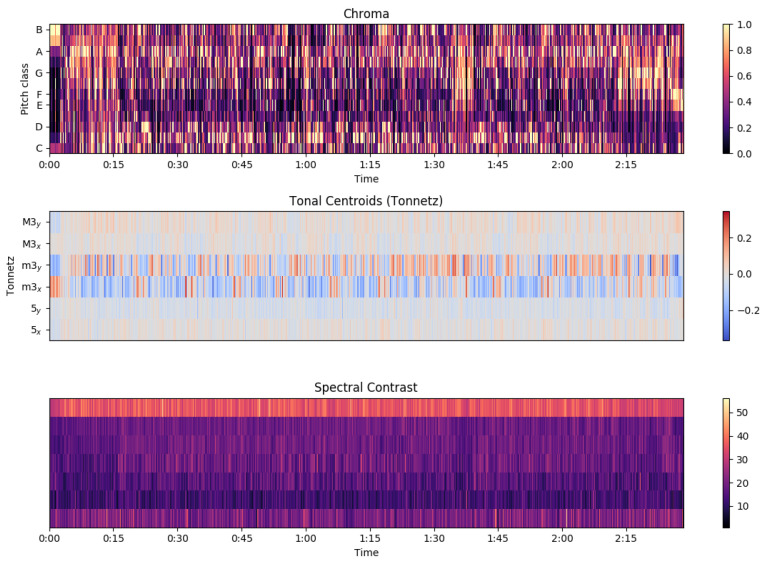
Chroma, Tonal Centroids, Spectral Contrast.

**Figure 3 sensors-22-01065-f003:**
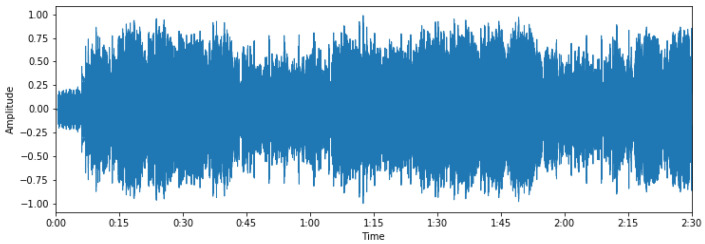
Audio signal of a musical track.

**Figure 4 sensors-22-01065-f004:**
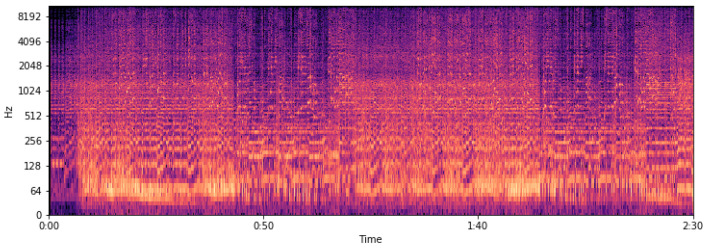
Spectrogram of a musical track.

**Figure 5 sensors-22-01065-f005:**
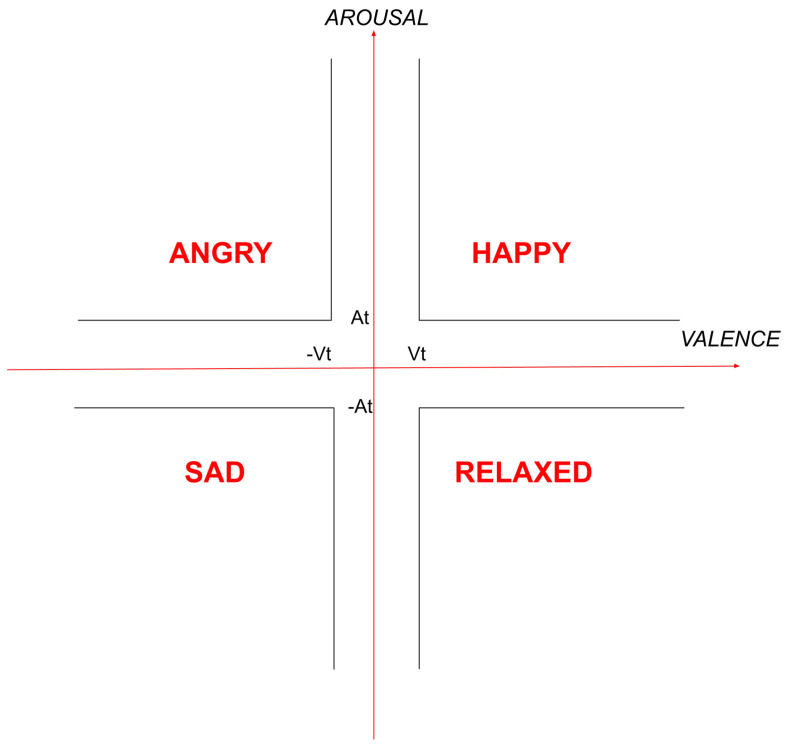
Discrimination of four mood classes in Circumplex model [18].

**Figure 6 sensors-22-01065-f006:**
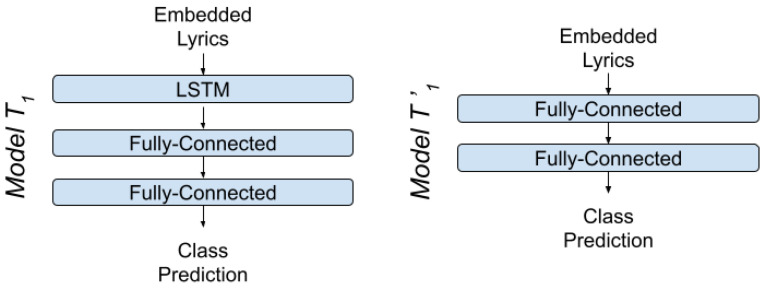
Architecture of Models $T_{1},T_{1}^{\prime }$.

**Figure 7 sensors-22-01065-f007:**
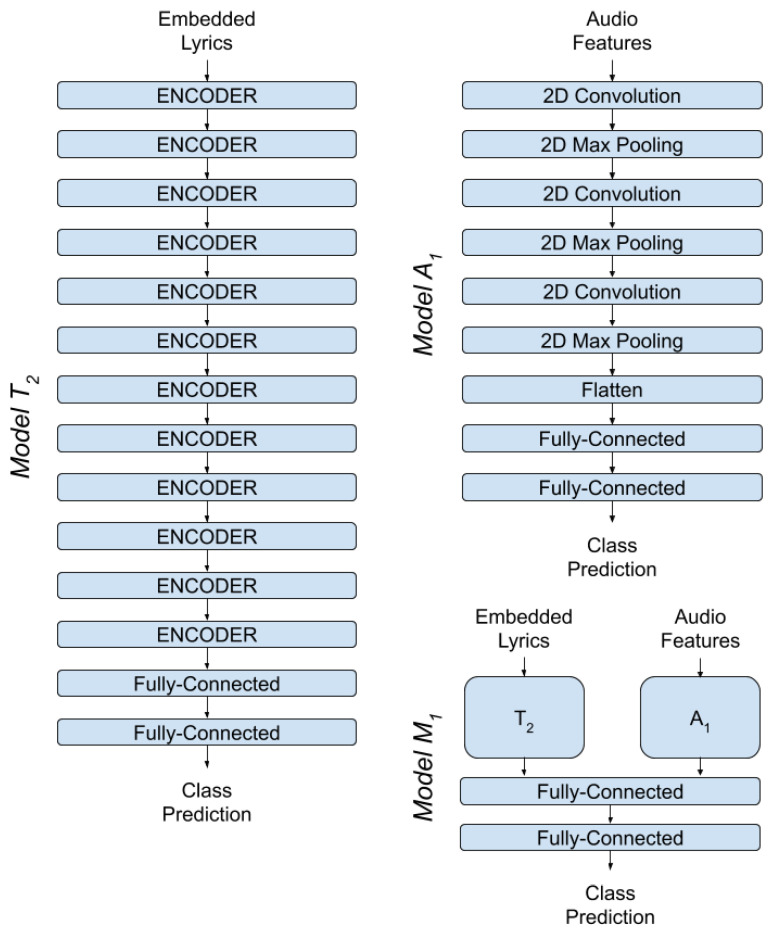
Architecture of Models $T_{2},A_{1},M_{1}$.

**Figure 8 sensors-22-01065-f008:**
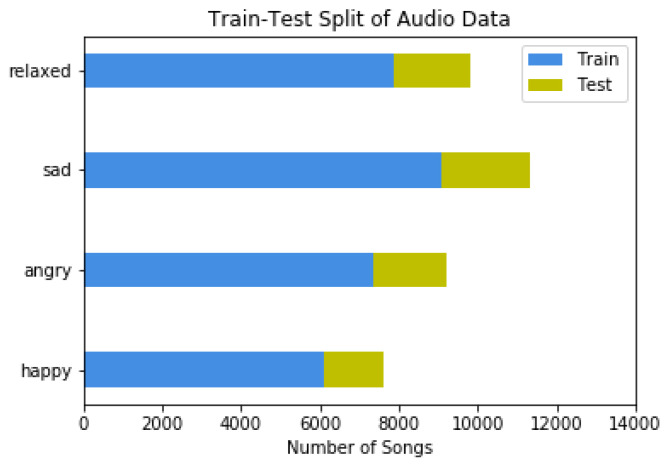
Train-Test spit of Audio Data.

**Figure 9 sensors-22-01065-f009:**
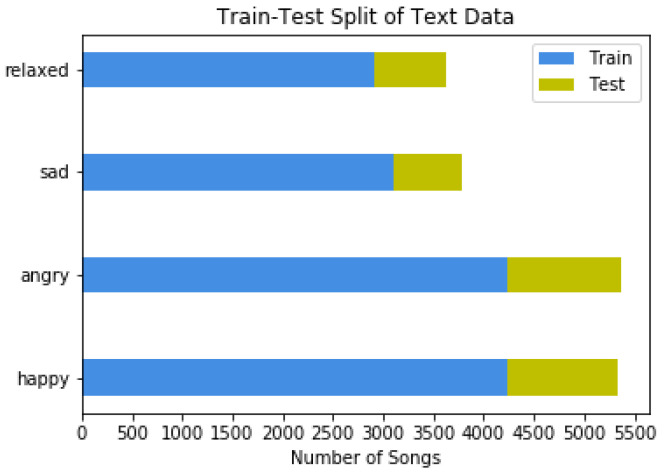
Train-Test spit of Text Data.

**Figure 10 sensors-22-01065-f010:**
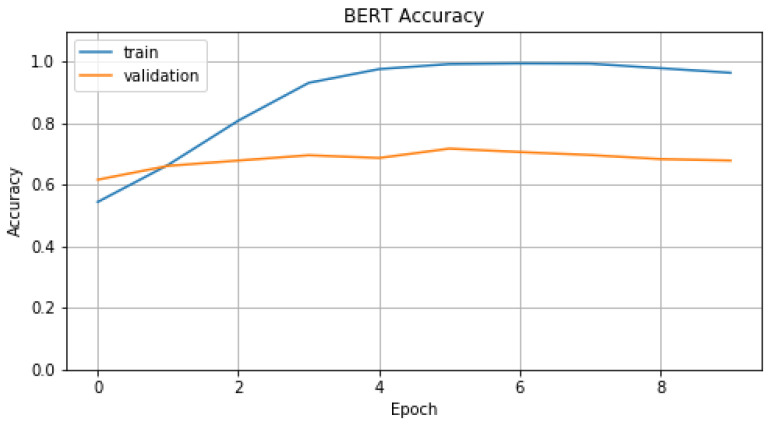
Accuracy of $T_{2}$.

**Figure 11 sensors-22-01065-f011:**
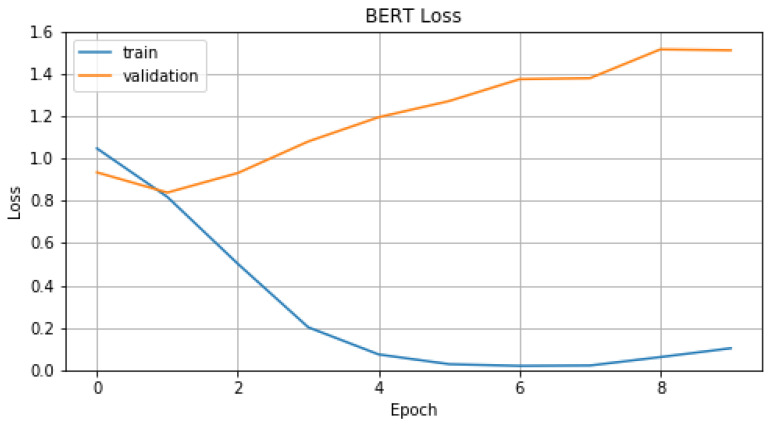
Loss of $T_{2}$.

**Figure 12 sensors-22-01065-f012:**
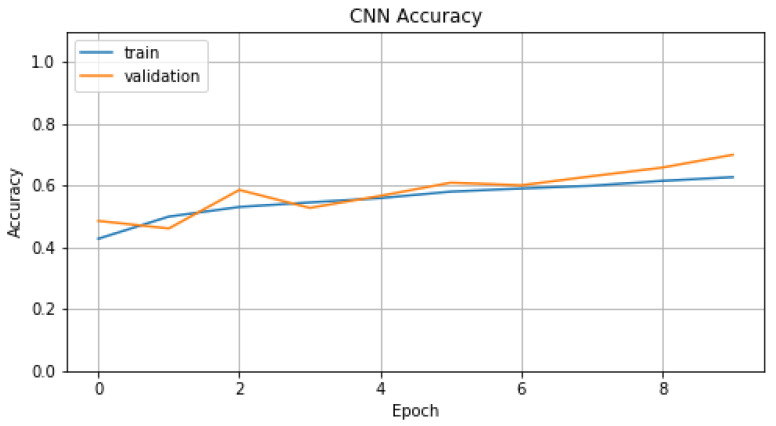
Accuracy of $A_{1}$.

**Figure 13 sensors-22-01065-f013:**
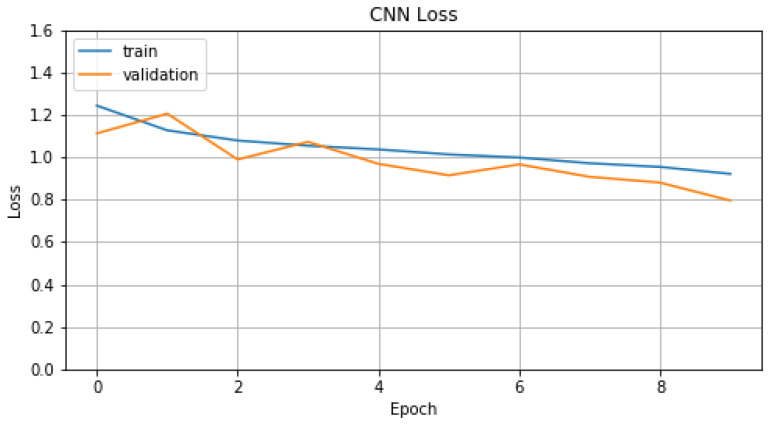
Loss of $A_{1}$.

**Figure 14 sensors-22-01065-f014:**
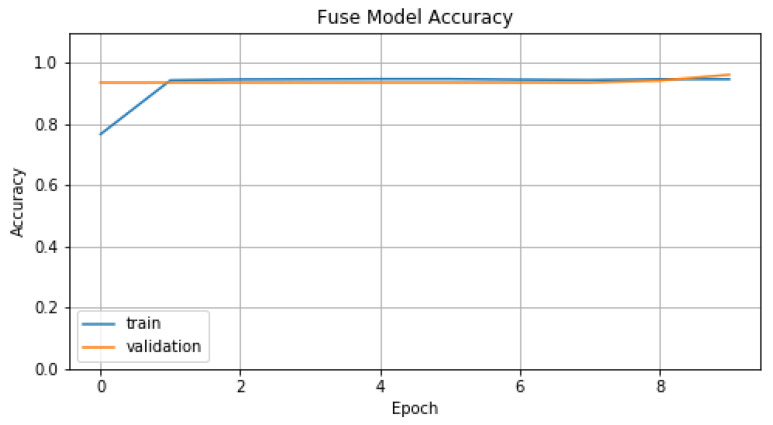
Accuracy of $M_{1}$.

**Figure 15 sensors-22-01065-f015:**
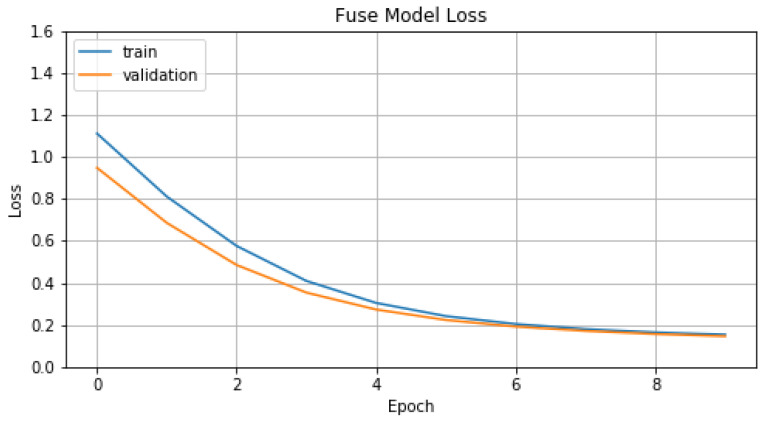
Loss of $M_{1}$.

**Table 1 sensors-22-01065-t001:** Association between structural features of music and emotion [23].

Structural Feature	Definition	Associated Emotion
Tempo	The speed or pace of a musical piece	Fast tempo: happiness, excitement, anger. Slow tempo: sadness, serenity.
Mode	The type of scale	Major tonality: happiness, joy. Minor tonality: sadness.
Loudness	The physical strength and amplitude of a sound	Intensity, power, or anger
Melody	The linear succession of musical tones that the listener perceives as a single entity	Complementing harmonies: happiness, relaxation, serenity. Clashing harmonies: excitement, anger, unpleasantness.
Rhythm	The regularly recurring pattern or beat of a song	Smooth/consistent rhythm: happiness, peace. Rough/irregular rhythm: amusement, uneasiness. Varied rhythm: joy.

**Table 2 sensors-22-01065-t002:** Song Classification.

Valence (V) and Arousal (A) Values	Mood
$A>A_{t}$ and $V>V_{t}$	Happy
$A>A_{t}$ and $V<-V_{t}$	Angry
$A<-A_{t}$ and $V<-V_{t}$	Sad
$A<-A_{t}$ and $V>V_{t}$	Relaxed

**Table 3 sensors-22-01065-t003:** Evaluation values of $T_{1}$ and $T_{2}$.

Model	Embedding Method	Loss	Accuracy (%)
$T_{1}^{{}^{\prime }}$	BoW	1.287	65.49
$T_{1}^{{}^{\prime }}$	TF-IDF	1.381	67.98
$T_{1}$	Word2Vec	1.262	41.66
$T_{1}$	GloVe	1.064	53.33
$T_{2}$	Bert	1.353	69.11

**Table 4 sensors-22-01065-t004:** Evaluation values of feature matrices.

Feature Combination	Accuracy (%)
Mel	64.97
Mel, Log-Mel	68.38
Mel, Chroma, Tonnetz, Spectral Contrast	60.86
Log-Mel, Chroma, Tonnetz, Spectral Contrast	58.96
MFCC, Chroma, Tonnetz, Spectral Contrast	65.36
Mel, Log-Mel, MFCC, Chroma, Tonnetz	69.77
Mel, Log-Mel, MFCC, Chroma, Tonnetz, Spectral Contrast	70.34

**Table 5 sensors-22-01065-t005:** Evaluation values of $T_{1}$, $T_{2}$ and $A_{1}$.

Model	Loss	Accuracy (%)
$T_{1}^{{}^{\prime }}$	1.381	67.98
$T_{2}$	1.353	69.11
$A_{1}$	0.743	70.51

**Table 6 sensors-22-01065-t006:** Evaluation values of $M_{1}$, $A_{1}$, $T_{1}$ and $T_{2}$.

Model	Loss	Accuracy (%)	Computational Time
$T_{1}^{{}^{\prime }}$	1.381	67.98	0 m 25.391 s
$T_{2}$	1.353	69.11	18 m 12.444 s
$A_{1}$	0.743	70.51	80 m 13.064 s
$M_{1}$	0.156	94.58	3 m 38.551 s

**Table 7 sensors-22-01065-t007:** Comparison of our model with other research papers that use MoodyLyrics [18].

Model	Accuracy (%)
XLNet [12]	94.78
BiLSTM [46]	91.08
NgramCNN [47]	75.60
XGBoost [48]	89.91
CNN [49]	72.24
BERT+CNN(ours)	94.58

## Data Availability

MoodyLyrics Dataset: https://dl.acm.org/doi/10.1145/3059336.3059340 (accessed on 21 January 2022).

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
