# Peer review of "Multi-Modal Song Mood Detection with Deep Learning†"

_sensors, 2022, doi:10.3390/s22031065_

Round 1

Reviewer 1 Report

The manuscript proposed a deep learning-based mixed model for song mood detection. The authors compare the prediction result among lyrics only, audio-only, lyrics + audio. The result shows proved the hypothesis that more diverse information will increase the accuracy. Based on this content, I have the following suggestion:
 1), Could you explain more about the difference and uniqueness of song (lyrics, audio)from NLP(words, audio?
2), Could you compare your result with other researchers based on accuracy or novelty in your models? 

Author Response

Response to Reviewer 1 Comments

The manuscript proposed a deep learning-based mixed model for song mood detection. The authors compare the prediction result among lyrics only, audio-only, lyrics + audio. The result shows proved the hypothesis that more diverse information will increase the accuracy. Based on this content, I have the following suggestion:

Point 1: Could you explain more about the difference and uniqueness of song (lyrics, audio)from NLP(words, audio) ?

Response 1: We complemented two new paragraphs for this point. The first paragraph at the end of Chapter 2 (line 84) , where we discuss about the task of Emotion Recognition and how the available data can help towards the direction of the task. The second paragraph is the paragraph right before section 2.2.1 (line 165), here we discus a little more about how NLP applies to the task and how we are going to develop our solutions.

Point 2: Could you compare your result with other researchers based on accuracy or novelty in your models?

Response 2: For this point we further expanded introduction, especially the last paragraph showcases the novelty of our work (line 62), and also we added Table 7 where we compare our results with other papers on the dataset we worked on (MoodyLyrics).

(Line numbering refers to the pdf line numbering not the Latex )

Reviewer 2 Report

An interesting paper focused on signal processing, including MIR and CNNs. However, at first glance, the Title is far too short, it is not informative and does not reflect the contents of the manuscript. Consider modifying and extending it.

The introduction is informative, however it is far too short. Where is the review of previously published papers, advancements in MIR, speech/music analysis and NLP, CNN, deep learning and AI applications, perception of signals and the effect of the background of individuals on their music taste, preferences, etc.? All of those need to be addressed and followed by appropriate citations.

Moreover, what is the added value of this paper? Is it yet another paper on MIR + CNN or does it bring some novel and significant findings? It should be clearly stated and highlighted.

About the utilized sourced dataset – do mention about the genre of those songs, file format, sampling frequency, bitrate, channel mode, length of the file, etc. Additionally, why did you chose this particular database?

Figs.1 and 2 – what music piece is analyzed? What does it reflect? What is its genre, title, is it only vocal or vocal + instrumental (and what instruments)? Is it available in an uncompressed or compressed audio file, etc. Is it a mix or an unprocessed file? Additional information are required.

Equations and mathematical formulas seem plausible and free of error. All of them are properly edited and enumerated.

Fig. 3 could have larger fonts, so that it is more legible. It would also be advisable to include it in higher resolution and/or different file format.

Dataset Preparation – additional information is required about the handling (processing) of audio files. Were the input (original) files the same as the output (processed) ones?

What about the hardware (laboratory stand and its specification) and software (utilized programs, libraries, third party resources) layer utilized during this study. Was this analysis carried out in real time or non real time? Did the Authors utilized Matlab or Octave?

Figs. 4, 5, 6, 7, 8 could be of higher resolution and or different file format – they seem a bit blurred.

If Authors, as Engineers, present charts/plots depicting Accuracy, Loss, etc. – those axes should be uniformly formatted. In the current form, the Y axis differs (the range of min-max values) from figure to figure.

Multiple editorial and formatting issues are present, e.g. lack of space or multiple space signs between subsequent words, punctuation, etc.

The Discussion chapter is far too short. Surely, as researchers, academics and professionals working in the field for a couple of years, you can provide more feedback, not to mention inspiration for the potential reader.

An additional Future Works chapter seems necessary. Mention about open questions and aspects that still require further investigation.

Taking all the above into consideration, the paper requires a major revision. Authors are encouraged to extend, modify and upgrade their manuscript.

Author Response

Response to Reviewer 2 Comments

Point 1: An interesting paper focused on signal processing, including MIR and CNNs. However, at first glance, the Title is far too short, it is not informative and does not reflect the contents of the manuscript. Consider modifying and extending it.

Response 1: We changed the Title of ou manuscript to Multi-modal song mood detection with Deep Learning (CCN+BERT).

Point 2: The introduction is informative, however it is far too short. Where is the review of previously published papers, advancements in MIR, speech/music analysis and NLP, CNN, deep learning and AI applications, perception of signals and the effect of the background of individuals on their music taste, preferences, etc.? All of those need to be addressed and followed by appropriate citations.

Response 2: Introduction was highly updated according to your useful points (lines 23-27, 28-37, 46-61).

Point 3: Moreover, what is the added value of this paper? Is it yet another paper on MIR + CNN or does it bring some novel and significant findings? It should be clearly stated and highlighted.

Response 3: The contributions of our work was clarified and complemented at the end of introduction (lines 70-74).

Point 4: About the utilized sourced dataset – do mention about the genre of those songs, file format, sampling frequency, bitrate, channel mode, length of the file, etc. Additionally, why did you chose this particular database?

Response 4: In lines 308-314 we completed the reason we chose MoodyLyrics as the dataset we experimented with. Also, in the first paragraph of section 3.2 (lines 398-407) we showcase the technical features of the utilized dataset.

Point 5: Figs.1 and 2 – what music piece is analyzed? What does it reflect? What is its genre, title, is it only vocal or vocal + instrumental (and what instruments)? Is it available in an uncompressed or compressed audio file, etc. Is it a mix or an unprocessed file? Additional information are required.

Response 5: Figs. 1 and 2 are replaced and the depicted music piece is described in lines 143-150. We also added Figs. 3, 4 which are the representations of the extracted features from audio signal.

Point 6: Equations and mathematical formulas seem plausible and free of error. All of them are properly edited and enumerated.

Response 6: Thank you.

Point 7: Fig. 3 could have larger fonts, so that it is more legible. It would also be advisable to include it in higher resolution and/or different file format.

Response 7: We tried to fix that issue but could not get much better results. Fig. 3 is now fig. 5.

Point 8: Dataset Preparation – additional information is required about the handling (processing) of audio files. Were the input (original) files the same as the output (processed) ones?

Response 8: Lines 413-414 were added to clarify the difference of original and processed files.

Point 9: What about the hardware (laboratory stand and its specification) and software (utilized programs, libraries, third party resources) layer utilized during this study. Was this analysis carried out in real time or non real time? Did the Authors utilized Matlab or Octave?

Response 9: A paragraph were added at the end of chapter 5 (lines 638-643), to provide technical details about the hardware and software utilized in our work.

Point 10: Figs. 4, 5, 6, 7, 8 could be of higher resolution and or different file format – they seem a bit blurred.

Response 10: Figs. 4-8 are now figs. 6-10. We tryied our best but overleaf seem to be compressing them a little which results in the poor quality. To address that we could send our figs. separately to editor. Also fig. 6 is replaced.

Point 11: If Authors, as Engineers, present charts/plots depicting Accuracy, Loss, etc. – those axes should be uniformly formatted. In the current form, the Y axis differs (the range of min-max values) from figure to figure.

Response 11: We changed figs. 11-16 to uniform Y-axis, according to your proposal.

Point 12: Multiple editorial and formatting issues are present, e.g. lack of space or multiple space signs between subsequent words, punctuation, etc.

Response 12: We tried our best to correct this issues throughout the manuscript.

Point 13: The Discussion chapter is far too short. Surely, as researchers, academics and professionals working in the field for a couple of years, you can provide more feedback, not to mention inspiration for the potential reader.

Response 13: We expanded conclusion chapter 6 (lines 653-658. 659-670) with more feedback on our work.

Point 14: An additional Future Works chapter seems necessary. Mention about open questions and aspects that still require further investigation.

Response 14: Future works is added at the end of chapter 6 (Conclusion and Future work) in lines 671-689.

Point 15: Taking all the above into consideration, the paper requires a major revision. Authors are encouraged to extend, modify and upgrade their manuscript.

Response 15: We hope that our renewed manuscript addresses your useful points and will meet the journals standards.

(Line numbering refers to the pdf documents' line numbers)

Round 2

Reviewer 2 Report

Thank you for addressing to my suggestion and comments. The revised version of the manuscript is much better compared to the initial submission.